# Portal Vein Thrombosis in the Setting of Cirrhosis: A Comprehensive Review

**DOI:** 10.3390/jcm11216435

**Published:** 2022-10-30

**Authors:** Aitor Odriozola, Ángela Puente, Antonio Cuadrado, Coral Rivas, Ángela Anton, Francisco José González, Raúl Pellón, Emilio Fábrega, Javier Crespo, José Ignacio Fortea

**Affiliations:** 1Gastroenterology and Hepatology Department, Clinical and Translational Research in Digestive Diseases, Valdecilla Research Institute (IDIVAL), Marqués de Valdecilla University Hospital, 39008 Santander, Spain; 2Radiology Department, Marqués de Valdecilla University Hospital, 39008 Santander, Spain

**Keywords:** portal hypertension, cirrhosis, portal vein thrombosis, anticoagulation, transjugular intrahepatic portosystemic shunt

## Abstract

Portal vein thrombosis constitutes the most common thrombotic event in patients with cirrhosis, with increased rates in the setting of advanced liver disease. Despite being a well-known complication of cirrhosis, the contribution of portal vein thrombosis to hepatic decompensation and overall mortality is still a matter of debate. The incorporation of direct oral anticoagulants and new radiological techniques for portal vein recanalization have expanded our therapeutic arsenal. However, the lack of large prospective observational studies and randomized trials explain the heterogenous diagnostic and therapeutic recommendations of current guidelines. This article seeks to make a comprehensive review of the pathophysiology, clinical features, diagnosis, and treatment of portal vein thrombosis in patients with cirrhosis.

## 1. Introduction

Portal vein obstruction can occur due to a malignant tumor (frequently but improperly referred to as malignant thrombosis), in which the obstruction is secondary to portal vein narrowing, and/or direct invasion of the portal vein by the neoplasm, or due to non-malignant portal vein thrombosis (PVT). The latter is defined as a thrombus that develops within the portal vein trunk and intrahepatic portal branches, which may also involve the splenic (SV) or superior mesenteric veins (SMV). In the absence of recanalization, the portal venous lumen is obliterated, and porto-portal collaterals develop, resulting in portal cavernoma. Non-malignant PVT can arise in two settings depending on the presence/absence of cirrhosis. This differentiation is critical since etiology, manifestations, natural history, and therapeutic options differ [1,2,3,4]. Previous early literature combined patients in these two settings and this should be accounted for to better interpret the results [5,6,7,8].

This article seeks to make a comprehensive review of the pathophysiology, clinical features, diagnosis, and treatment of PVT in patients with cirrhosis. It constitutes the most common thrombotic event in this population, with increased rates in the setting of advanced liver disease. Despite being a well-known complication of cirrhosis, the contribution of PVT to hepatic decompensation and overall mortality is still a matter of debate. There is consequently no consensus on its optimal management, and no definitive recommendations were reported in clinical guidelines or consensus conferences [1,2,3,4,9,10,11,12].

## 2. Anatomy of the Venous Portal System

The liver is a highly vascular organ that receives up to 25% of the total cardiac output from a dual blood supply. The hepatic artery delivers well-oxygenated blood and comprises approximately 25% of total hepatic blood flow, whereas the remaining 75% is deoxygenated blood supplied by the portal vein [13]. These two afferent vascular systems drain into the modified capillary network of the liver, composed of fenestrated sinusoids. From the sinusoids, blood flows into the central veins that drain in the hepatic veins and then in the inferior vena cava [2].

The hepatic portal system is the most known portal venous system in the body, hence its name. The latter is defined as a circulatory system in which veins connect two capillary beds without first carrying blood to the heart. In the case of the hepatic portal venous system, capillary blood from the entire gastrointestinal tract (except for the upper esophagus and distal rectum), pancreas, gallbladder, and spleen is carried to the hepatic sinusoids. The portal vein is an 8-cm, valveless conduit originating from the confluence of the SMV and SV posterior to the neck of the pancreas [2]. The inferior mesenteric vein (IMV) is a tributary vessel that usually drains in the SV (up to 40% of the cases). However, there are many variants in the drainage of this vein [14]. In the other two most frequent variants, the IMV enters into the angle of the confluence of the SV and the SMV (30%) or the SMV (20%). Other lesser frequent variants (10%) consist of accessory mesenteric veins entering the SV or SMV.

The portal vein develops during the second and third month of gestation from two vitelline veins, which drain the yolk sac. These veins form few anastomoses between each other which result in the formation of the portal vein. Deviations from the normal process of these anastomosis result in variations in the branching pattern of the portal vein. Its distribution within the liver is segmental and closely follows the hepatic artery. Cheng Y et al. classified these intrahepatic portal vein variations into five different types (Figure 1) [15].

## 3. Classification

Terminology and classification systems of PVT vary extensively in the literature, with most developed exclusively in the liver transplant (LT) population [3]. Four of the major published PVT classification systems are presented in Table 1 [16,17,18,19]. More recently, the latest guidelines of the American Association for the Study of the Liver (AASLD) on vascular disorders and the Baveno VII workshop have promoted the establishment of a standardized terminology in describing PVT to allow comparison and external validation of future studies. They proposed a systematic documentation of initial site, extent/degree of luminal obstruction, and chronicity of PVT to enable subsequent evaluation of the spontaneous course and/or response to treatment (Figure 2) [3,4].

In line with the above, the initial description of PVT must detail the location and extension of the thrombosis, specifying whether it involves the intrahepatic branches, main portal vein, SV, and/or SMV. The percent of lumen occluded in each of these locations is graded as completely occlusive (no persistent lumen), partially occlusive (clot obstructing >50% of original vessel lumen), or minimally occlusive (clot obstructing <50% of original vessel lumen). If present, cavernous transformation (i.e., gross porto-portal collaterals without original portal vein seen) must be reported. This documentation is not only important for therapeutic decisions and evaluating response to treatment but also for establishing a correlation between the site of thrombosis and the clinical presentation of PVT. Indeed, the involvement of SMV may cause intestinal ischemia and development of ectopic varices, whereas SV thrombosis could lead to the appearance of fundic varices [3,4].

As far as the time course of thrombosis is concerned, PVT is classified as recent if PVT is presumed to be present for less than 6 months and as chronic if PVT persists longer than this time frame. The term recent is preferred over acute because the latter implies clinical symptoms and PVT is often asymptomatic and incidentally diagnosed. That is why precise dating is impossible in a significant proportion of patients. The 6-month threshold was chosen based on data from a prospective study of 102 patients with recent non-cirrhotic PVT. In this study, failure to recanalize within 6 months of PVT diagnosis led to development of cavernous transformation in most patients despite continued anticoagulation [20]. Similar results were observed in the setting of liver cirrhosis [21,22]. When chronic obstruction persists and cavernous transformation occurs, the latter term is preferred. However, cavernous transformation is not synonymous with chronic PVT since this finding can be demonstrated within 1–3 weeks after PVT onset [23].

Finally, depending on the spontaneous course and/or response to treatment, PVT is classified as progressive (thrombus increases in size or becomes completely occlusive), stable (no changes in size nor occlusion) or regressive (thrombus decreases in size or degree of occlusion) [3,4].

## 4. Pathophysiology and Risk Factors

As with any other thrombus, the pathogenesis of PVT is generally multifactorial and it is believed to be primarily determined by the interplay of the three physiological factors of Virchow’s triad: slow blood flow, hypercoagulability, and endothelial damage. Nevertheless, the exact contribution of each of these factors to PVT development was not fully elucidated [24].

### 4.1. Reduced Blood Flow

A reduced portal flow velocity secondary to a “steal effect” produced by porto-collateral circulation, combined with the increase in portal vein diameter commonly seen in patients with clinically significant portal hypertension (CSPH) seems to be the most important factor for PVT development in patients with cirrhosis. Specifically, a 15 cm/s threshold was proposed to identify patients at higher risk of developing PVT. This was first reported by Zocco et al. in a prospective study of 100 cirrhotic patients followed up for one year [25]. Since then, other retrospective and prospective studies have confirmed this finding [26,27,28]. However, a large satellite study of a prospective trial of ultrasound screening for hepatocellular carcinoma (HCC) did not and raised concerns regarding the reproducibility of portal vein blood flow velocity measurements, thus, making uncertain the generalization of specific thresholds [29]. Further indirect data in this regard comes from studies identifying a higher portal vein diameter and the presence of large porto-collateral vessels as risk factors for developing PVT [26,30,31]. The efficacy of a transjugular intrahepatic portosystemic shunt (TIPS) in restoring PVT patency by presumably increasing portal flow also supports this major role of altered portal hemodynamics in PVT development [32].

As nonselective β-blockers (NSBB) may reduce portal venous flow by decreasing cardiac output and by inducing splanchnic arterial vasoconstriction, it was suggested that they might increase the risk of PVT [33,34]. This is of great importance since indications of NSBB have recently widened, and they are now recommended not only for primary or secondary prevention of variceal bleeding but also to prevent decompensation in patients with compensated advanced chronic liver disease and CSPH [4]. A recent systemic review and meta-analysis of mostly retrospective studies showed an increased risk of PVT in subjects on NSSBs treatment (OR 4.62, 95% CI 2.50–8.53; *p* < 0.00001) [35]. However, there was high heterogeneity in relation to sample size, definition of PVT and stage of liver disease and most of the studies did not evaluate the potential role of confounding factors. Indeed, in two large prospective studies, NSBB were not associated with an increased risk of PVT after adjustment for variables related to the severity of portal hypertension [28,29]. Therefore, the use of NSBB should not be limited based on this concern, even more so considering their larger benefits in patients with cirrhosis [24].

### 4.2. Alterations in Coagulation

Cirrhosis has long been perceived as an acquired bleeding disorder as a consequence of thrombocytopenia and abnormal routine coagulation tests. However, unlike hereditary coagulopathies, cirrhosis affects the whole spectrum of the coagulation cascade (i.e., both procoagulant and anticoagulant factors) and is associated with both platelet hyperactivity and increased levels of von Willebrand factor, all of which results in a “rebalanced hemostasis”. This new equilibrium is fragile and can easily be tipped towards either a prohemorrhagic or a prothrombotic phenotype. Furthermore, it was suggested that this potential prothrombotic state is intimately involved in the progression of liver disease and could also be involved in PVT development [3,24,36,37].

Several studies have evaluated whether the hemostatic alterations associated with cirrhosis increase the risk of PVT and were extensively reviewed elsewhere [24]. Traditional coagulation tests (e.g., prothrombin time or partial thromboplastin time) do not adequately reflect this new hemostatic balance since they do not take into account the inhibition of thrombin by anticoagulant factors [3,36]. The ratio between FVIII and protein C has long been suggested to reflect this increase in coagulation potential [38], although recent data have challenged this assumption showing that despite predicting the development of complications of cirrhosis, it is unrelated to the coagulation status of patients with cirrhosis [39]. Regarding its role in PVT, contradictory results were published [28,40,41]. Similar findings were observed with other cirrhosis-related hemostatic alterations, such as the ratio of factor II to protein C, levels of coagulation factors, thrombomodulin resistance, fibrinolysis markers, plasminogen activator inhibitor-1 levels [42], or viscoelastic parameters [24]. Of note, the aforementioned studies evaluated coagulation factors in systemic blood with few studies evaluating whether the portal vein may represent a hypercoagulable vascular bed. Although initial studies described the existence of a relative hypercoagulability in this territory [43,44], a more recent study failed to replicate this finding in cirrhotic patients who underwent TIPS placement [45]. Its hypothetical contribution in PVT development was not tested as none of these studies evaluated patients with PVT.

There are fewer data on the role of platelet aggregation in PVT development. In opposition to previous studies suggesting platelet dysfunction in patients with cirrhosis, recent studies show that platelets are hyperfunctional in these patients, particularly in the decompensated stage and in the portal vein [46,47,48]. Moreover, this increased platelet aggregatory potential was associated with a higher risk of further decompensation, death, and PVT [47,49]. In line with these findings, a recent study showed that the ADAMTS-13/von Willebrand factor ratio was predictive of PVT [50]. These results establish a rational basis for evaluating the use of antiplatelet agents to prevent PVT and halt disease progression.

Other factors that may induce blood hypercoagulability in patients with cirrhosis are systemic inflammation, a well-recognized feature of decompensated cirrhosis [51], and HCC [52]. The former is scarcely studied with contradictory results [24,28]. A recent paper observed that serum albumin was inversely associated with PVT and suggested albumin as a modulator of the hemostatic system by reducing platelet activation through its inhibitory effects on oxidative stress. According to the authors, these findings established a rationale for randomized interventional studies to investigate the beneficial effects of albumin to prevent PVT in cirrhosis [53]. No information in this matter was provided in the long-term albumin administration trials [54,55]. As far as HCC is concerned, there is growing evidence suggesting that it is associated with pro-thrombotic alterations (i.e., increased platelet activation and function, enhanced thrombin generation, hypo-fibrinolysis, and elevated levels of prothrombotic microvesicles) that may synergistically contribute to hypercoagulability and thrombosis [52].

Regarding the role of inherited and other acquired prothrombotic disorders in PVT development, current data are conflicting. The limited number of studies available are mostly case-control studies with small sample sizes. Their study design, target population (diverse ethnicities and geographical locations), diagnostic criteria for PVT, and assessment of thrombophilic conditions vary widely and contribute to the inconsistent results [21,29,56,57,58,59,60,61,62,63,64,65,66,67,68,69,70,71,72,73,74,75,76,77]. Moreover, none of these studies have properly evaluated whether the presence of thrombophilia impacts the progression rate or response to treatment [77]. Among the different thrombophilic genetic defects, Factor V Leiden and prothrombin G20210A mutations are the most frequently studied. Three meta-analyses concluded that they increased the risk of PVT in patients with cirrhosis [78,79,80], although in one of them, this association was not shown for the prothrombin mutation [79], and all of them were biased by the quality of the studies included. Inherited protein C, protein S, or antithrombin III deficiencies are difficult to detect due to co-existent liver synthetic dysfunction [36]. Their levels, however, do not seem to be associated with PVT development [81]. The methylene tetrahydrofolate reductase C677T and plasminogen activator inhibitor—type 1 4G–4G mutations were also described as independent predictors of PVT [63,82], although these polymorphisms are not conclusively associated with increased thrombotic risk [82]. The role of other acquired prothrombotic disorders was less evaluated in patients with liver cirrhosis and PVT. In contrast to non-cirrhotic PVT, the relevance of myeloproliferative disorders and antiphospholipid syndrome is, so far, inconclusive [83]. Due to the conflicting data, current guidelines make no strong recommendations regarding testing for these conditions in either a screening capacity before PVT diagnosis or confirmatory once thrombosis has developed [1,2,3,4,10].

In summary, there is no solid evidence that hypercoagulability due to cirrhosis-related hemostatic alterations or to inherited and other acquired prothrombotic disorders plays a major role in the pathophysiology of PVT.

### 4.3. Endothelial Damage

Of the three components of Virchow’s triad, the hypothetical role of endothelial dysfunction in PVT generation is the least studied, partially due to the inaccessibility of the splanchnic territory (for review see [24]). Therefore, more studies are needed that compare endothelial-specific markers in blood from the portal area between cirrhotic patients with or without PVT.

Endothelial injury due to sclerotherapy, previous abdominal surgery, splenectomy, and portosystemic shunt surgery were also identified as risk factors for PVT, although altered portal venous blood flow due to some of these procedures also promotes thrombus formation [84].

## 5. Epidemiology

PVT constitutes the most common thrombotic event in the setting of cirrhosis. Defining its incidence and prevalence is difficult due to the heterogeneity of studies regarding the population included (LT candidates are the most studied), the definition of PVT, and the tests used for its diagnosis [3]. The few prospective studies have reported incidence rates in the range from 1.6% to 4.6% at 1 year [28,29,85,86]. A recent meta-analysis showed 1-year and 3-year cumulative incidences of 4.8% and 9.3%, respectively [87]. This incidence varies among compensated and decompensated patients. In the latter meta-analysis, the pool incidence of PVT was 9.9% in Child–Pugh class A and 18.3% in Child–Pugh class B–C. Higher rates were described in the presence of HCC (up to 40%) [84,88]. Incidence rates also vary by disease etiology with evidence that PVT is more frequently associated with nonalcoholic fatty liver disease [89].

## 6. Clinical Manifestations and Prognostic Impact

Diagnosis of PVT is most often asymptomatic and commonly discovered by routine imaging tests [1]. Symptoms ascribed to PVT are non-specific and include nausea, vomiting, mild abdominal pain, diarrhea, and loss of appetite. Rarely, mesenteric ischemia due to the extension of PVT to the SMV can occur [2]. Patients with more advanced liver cirrhosis are more protected than non-cirrhotic patients from this complication due to the decompression achieved through the frequent presence of porto-systemic collaterals [3]. It is, thus, important to establish a correlation between PVT features (time course, degree of occlusion, and stage of liver disease) and the clinical presentation [84]. For example, in the setting of acute abdominal pain, the finding of a partially occlusive PVT on ultrasound (US) should not defer the performance of a contrast-enhanced computed tomography (CT) or magnetic resonance (MR) scan to rule out other causes of abdominal pain (e.g., pileflebitis or malignant infiltration into the portal vein) and to better characterize the true extension of PVT.

In other instances, PVT is diagnosed in coincidence with a liver decompensation, and again, the temporal relationship should not be directly interpreted as evidence of causality. Indeed, the impact of PVT on the natural history and prognosis of cirrhosis is controversial, and whether PVT is merely a manifestation of progressive disease, or an actual cause of disease progression remains to be elucidated [3]. Discrepancies among studies regarding patient selection criteria (compensated vs. decompensated), degree and extent of thrombosis (occlusive vs. nonocclusive), treatment strategies (anticoagulation vs. no anticoagulation), sample size, and time of follow-up have led to conflicting data [90]. Hence, several prospective [29,85,91,92] and retrospective studies [30,93] have shown that PVT is not responsible for disease progression or increased mortality, whereas a randomized study by Villa et al. indirectly suggested the opposite. In this small, controlled trial, a 12-month course of 4000 IU/day enoxaparin in patients with Child B–C (7–10 points) cirrhosis not only prevented PVT but also improved survival and decompensation [94]. Moreover, PVT was shown to be independently associated with a higher risk of variceal bleeding and failure of endoscopic control of bleeding and rebleeding [95,96,97].

In LT recipients, the impact of PVT on survival after LT seems to depend on the size and extent of PVT at the time of surgery [98]. Two large transplant database analyses showed PVT as a strong independent predictive variable of posttransplant survival but did not stratify this risk according to the grade of PVT [99,100]. Other studies [16,101,102] and a meta-analysis [103] have shown that only a completely occlusive PVT increases post-transplant mortality. The threshold of PVT extension at which outcomes are worse is unknown but is probably related to the need for non-anatomical PV reconstructions (renoportal anastomosis, cavoportal hemitransposition, or portal vein arterialization). These cases have worse outcomes by adding technical difficulties and increasing graft ischemic times [19]. Unfortunately, there are no randomized controlled trials showing that PVT therapy improves post-transplant survival. Due to these discrepant results and lack of randomized controlled trials, guidelines differ on their recommendations. Hence, the AASLD guideline on vascular disorders states that there are insufficient data to recommend pretransplant treatment of PVT with the goal of improving posttransplant outcomes [3], while the Baveno VII consensus recommends anticoagulation in potential LT candidates independently of the degree of occlusion and extension with the goal of preventing re-thrombosis or progression of thrombosis to facilitate adequate portal anastomosis in LT and reduce post-transplant morbidity and mortality [4]. Similarly, some guidelines only recommend screening for PVT in potential LT candidates at the time of screening for HCC (1, 4, 10), while others do not make any specific recommendations [3].

## 7. Natural History

PVT is a heterogeneous condition also with respect to its natural history, and this heterogeneity makes PVT a unique entity among venous thromboses [24]. On the one hand, the spontaneous recanalization of PVT is extensively described [21,29,30,92,93,102,104]. A recent meta-analysis comparing anticoagulated vs. non-anticoagulated patients with cirrhotic PVT showed a rate of spontaneous portal vein recanalization of 42% [105]. The probability of this event is higher in compensated cirrhosis or with partial PVT (up to 70%) [29] and much lower in patients with decompensated cirrhosis and those listed for LT [21,98,102,106]. The different imaging techniques used to stage the extent of the thrombus also contribute to the heterogeneous rates of spontaneous recanalization or of progression reported. In addition to these factors, no other predictors of spontaneous improvement or progression are known, which makes it difficult to evaluate the real efficacy of the different treatments used for PVT. It must be noted that although spontaneous resolution of PVT may occur, especially if non-occlusive and in compensated cirrhosis, PVT progression occurs in 33% of untreated patients [105].

On the other hand, the rates of portal vein recanalization after anticoagulation are substantially lower than in other thromboses, especially in an aged thrombus [24]. Driever et al. recently proposed an explanation for this finding [107]. In their study, they described the composition and structure of nonmalignant cirrhotic PVT that were collected during LT in 79 patients. They observed that all PVT consisted of tunica intima thickening of the portal vein vessel wall in an appearance resembling intimal fibrosis, with only one-third of the thrombi containing an additional fibrin-rich thrombus. Based on these findings, they suggested changing the name PVT to portal vein stenosis and that the absence of fibrin in most patients may explain the low rates of recanalization with anticoagulant therapy.

## 8. Diagnosis

The initial diagnosis of PVT is often made with Doppler US and this diagnostic technique is the screening method of choice for PVT [2,4]. Doppler US may demonstrate hyperechoic material within the vessel lumen, dilatation of the portal vein, and diminished portal venous flow [108,109,110]. US has a sensitivity ranging from 73–93%, specificity of 99%, and positive predictive value of 87–96% compared with angiogram [16] and CT scan [108]. Advantages of US Doppler over other diagnostic tests include lower cost, wider availability, and lack of radiation exposure. Despite being an excellent initial screening test, US Doppler is an operator-dependent exploration and has lower reliability in the presence of bowel gas, obesity, partially occlusive PVT, and in delimiting the extension of the thrombi to SV and SMV. Furthermore, it may be difficult to differentiate bland thrombi from malignant portal vein invasion. For all these reasons, current guidelines recommend performing a contrast-enhanced imaging study following PVT diagnosis with US [1,2,3,4,10]. Both CT and MR scans are excellent techniques for diagnosing PVT and portal cavernoma [108]. In comparison to CT, MR scan has the advantages of less radiation and a better safety profile, but it is limited by motion and flow artifacts, lower availability, higher cost, and technical difficulties in patients with implanted metallic devices or surgical clips [2]. Another less studied technique in this setting is endoscopic US. In a small cohort of patients with and without cirrhosis it had a sensitivity of 81% and a specificity of 93% for diagnosing PVT [111]. However, given its moderately invasive nature and inability to definitively assess for HCC or mesenteric infarction, it is not routinely recommended in the diagnostic algorithm of PVT.

The exclusion of tumoral invasion of the portal vein is essential to both decisions regarding further management of HCC and determination of LT candidacy [84]. It may occur in up to 12% to 20% of the patients with HCC [112,113]. Findings that support this malignant invasion include enlarged portal vein diameter, enhancement of the thrombus in the arterial phase of contrast injection, neovascularity, distance from tumor to thrombus lower than 2 cm, and tumor size > 5 cm [114,115]. The recently proposed A-VENA criteria incorporates all these findings, except for tumor size, and also includes alfa-fetoprotein > 1000 ng/dL. In patients with ≥3 of these criteria, the diagnosis of malignant invasion could be accurately made (100% sensibility, 94% specificity, 80% positive predictive value and 100% negative predictive value) [116].

In regard to the “age” of PVT, features of recent PVT include hypoechogenic and hypodense thrombus, increased attenuation in the portal vein on an unenhanced CT scan, and increased attenuation in the portal vein and a central lucency on a contrast-enhanced CT scan. Associated hepatic perfusion changes can also be seen in the form of increased hepatic parenchymal enhancement in the arterial phase and reduced enhancement during the portal phase. In contrast, calcification within the wall of a thrombus and presence of cavernoma suggest chronicity [2,84] (Figure 3). It must be pointed out that the former can only be detected using US or CT but not MR and that cavernous transformation can occur within 1–3 weeks after PVT onset [23] (Figure 4).

## 9. Prophylaxis

There is only one trial assessing the use of anticoagulation for preventing PVT. In this study by Villa et al., 70 patients with cirrhosis (Child B7-C10) were randomized to receive enoxaparin at prophylactic doses (4000 IU/day for 48 weeks) vs. no treatment. Patients receiving enoxaparin had lower incidence of PVT and a sustained decrease in decompensation events that exceeded the expected from the reduction in PVT. The authors hinted at an enoxaparin-induced improvement of intestinal barrier function and reduced bacterial translocation as a potential mechanism [94]. It was also postulated that these beneficial effects could also be related to the antifibrotic effects of anticoagulation. Indeed, increasing evidence suggests that the potential prothrombotic state associated with cirrhosis leads to liver fibrosis development and progression in liver disease presumably by generating thrombi in the hepatic microcirculation that cause parenchymal extinction and by activating hepatic stellate cells through thrombin and Factor Xa via protease-activated receptors [37,117]. Results from a confirmatory study using ribaroxaban instead of enoxaparin are eagerly awaited (ClinicalTrials.gov Identifier: NCT02643212).

## 10. Treatment

The decision to treat a PVT in the setting of cirrhosis is determined by the age and extent of thrombus, the presence of symptoms, and the patient’s transplant status [84]. In patients with concern for intestinal ischemia, early initiation of anticoagulation and immediate consultation with surgery, critical care, interventional radiology, and hematology is advised [3]. In the patient without ischemic symptoms, the weak existing evidence on the beneficial effects of therapy and the uncertain impact of PVT on the natural history of cirrhosis explain the heterogenous recommendations of current guidelines. In this scenario, the aim of treatment is to prevent clot extension that could potentially lead to progression of portal hypertension, hinder a future LT, or preclude a conventional end-to-end portal vein anastomosis, thus, reducing post-transplant morbidity and mortality [3,4]. Overall, and especially in non-LT candidates, treatment should be considered on a case-by-case basis. Table 2 shows the recommendations of the different guidelines regarding treatment indications and other issues.

### 10.1. No Treatment

A conservative approach is generally considered in asymptomatic non-LT candidates in whom the uncertainties of the survival benefit of anticoagulation are greater. In this scenario, recent guidelines suggest a conservative approach in thrombosis of small intrahepatic sub-branches of the portal vein or minimally occlusive (<50% obstruction of the lumen) PVT. In case of progression at serial imaging, anticoagulation should then be considered [3,4].

The latter strategy can be also applied to patients with chronic complete occlusion of PVT or cavernous transformation of the portal vein with established collaterals in whom there is no established benefit for anticoagulant therapy [3]. The main objective of anticoagulation in these patients is to prevent recurrent thrombosis and, to a lesser extent, recanalization. The decision-making process in this setting should be based on bleeding risk, SMV involvement, presence of a thrombophilic condition, LT candidacy, and patient preferences. Thus, the guideline on disorders of the hepatic and mesenteric circulation of the American College of Gastroenterology suggests anticoagulation in patients with chronic PVT only if there is (i) evidence of inherited thrombophilia, (ii) progression of thrombus, (iii) history of bowel ischemia due to thrombus extension into the mesenteric veins, or (iv) PVT in a patient awaiting LT [2].

### 10.2. Anticoagulation

Anticoagulation is the mainstay of PVT treatment. The majority of studies assessing the safety and effectiveness of anticoagulants in patients with cirrhosis are small retrospective studies that differ in inclusion criteria, magnitude of PVT (predominancy of partially occlusive PVT), timing and type of anticoagulation, and in the assessment of outcomes. Moreover, anticoagulation decisions were based on provider’s judgment and were not based on predefined protocols [2]. With these limitations, data from these studies [21,22,68,73,77,102,104,118,119,120,121,122,123,124,125,126] and from aggregate-data meta-analyses [105,127,128,129] have shown that anticoagulation is safe and effective in achieving portal vein recanalization, but its effect on survival is still uncertain.

In regard to effectiveness, the latter meta-analyses have shown rates of recanalization ranging between 66.6% and 71.5% for any degree of recanalization and between 40.8% and 53% for complete portal vein recanalization. In contrast, rates of thrombus progression despite anticoagulation range between 5.7 and 9%. Early administration of anticoagulation (<14 days in one study [119] or <6 months in others [21,120,130]) is of utmost importance for treatment success with the mean time of recanalization ranging from 5.5 to 8 months. However, delayed responses even after 1 year of treatment were reported [68]. Other factors that are associated with a good response are less advanced disease, less extensive thrombosis [119,126], degree of SMV occlusion less than 50%, lower platelet count, absence of previous PHT-related bleeding, and lower spleen thickness at baseline [129,131]. Importantly, when anticoagulation is stopped, PVT relapses in 30–40% of the patients [118,119,132]. The time elapsed between discontinuation of anticoagulation therapy and recurrence of thrombosis ranges between 2 and 5 months [119,132]. All these findings explain the recommendations from current guidelines to anticoagulate for at least 6 months and to maintain anticoagulation until LT (Table 2). The lack of evidence to recommend a specific timing or type of imaging study regardless of the decision to initiate anticoagulation or not is also of note. The AASLD suggests serial imaging every 2 to 3 months to assess treatment response and every 3 months if a conservative approach is chosen, but anticoagulation is considered in case of PVT progression.

The former recanalization rates in patients treated with anticoagulants are significantly higher than those seen in untreated patients (25–42% for overall recanalization with odds ratio ranging between 2.61 and 4.8 and 33% for complete portal vein recanalization with odds ratio ranging between 2.14 and 3.4). Similarly, PVT progression is more likely in untreated patients (33% with odds ratio ranging between 0.06 and 0.26) [105,127,128,129]. The impact of these higher rates of recanalization under anticoagulation therapy is still uncertain, with two meta-analyses showing a decreased risk of variceal bleeding [105,128] and another an improvement in overall survival [129]. An unpublished individual patient data meta-analysis also showed an increased overall survival and found that the beneficial effect of anticoagulation largely depended on portal vein recanalization [133].

Whatever the impact, anticoagulant treatment was shown to be safe, with similar major and minor bleedings rates when compared to untreated patients (10.3–11%) [105,129]. Moreover, anticoagulation was not associated with an increase in 5-day treatment failure or 6-week mortality in patients with cirrhosis having an episode of upper gastrointestinal bleeding [134]. A platelet count below 50 × 109/L was identified as a risk factor for bleeding from any site in patients with cirrhosis and PVT receiving anticoagulation [119]. These patients are also at higher risk of PVT, and consequently, management should be assessed on a case-by-case basis [4].

Current options for anticoagulation include vitamin K antagonists (VKAs), unfractionated heparin, low-molecular-weight heparin (LMWH), fondaparinux, and direct oral anticoagulants (DOACs). Table 3 shows the main characteristics of each therapeutic option. Most of the published studies have used VKAs and LMWH, and both seem to have similar effectiveness [105].

#### 10.2.1. Unfractionated Heparin

Treatment of PVT with unfractionated heparin may be used as the initial agent in the presence of renal insufficiency and/or in patients with concern for intestinal ischemia due to the possibility of rapid reversal of anticoagulation [2]. However, its intravenous administration precludes its use as a long-term treatment.

#### 10.2.2. Low Molecular Weight Heparin (LMWH)

LMWH is traditionally considered the initial treatment of choice for PVT. Due to its parenteral administration, patient compliance and quality of life can be compromised. Therefore, it is generally used as a “bridge” therapy in patients that will ultimately receive VKAs or DOACs. The duration of this “bridge” therapy varies greatly among centers, with some initiating LMWH and VKAs simultaneously (with discontinuation of LMWH once the INR reaches therapeutic range of 2–3) and others (similar to our center) postponing oral anticoagulation for a month. However, in some specific patients, LMWH may be more suitable than VKAs, such as in patients with refractory ascites requiring periodic paracentesis or patients with prolonged INR.

In terms of dose, a small study showed that enoxaparin 1 mg/kg twice daily had similar efficacy with fewer complications than 1.5 mg/kg daily [121]. Whether a reduction in dose is necessary in patients with high bleeding risk (e.g., severe thrombocytopenia) is unknown, with one small study showing that a reduced dose was not associated with a decrease in efficacy [130]. Measuring anti–Factor Xa activity to monitor the anticoagulant effect of LMWH may lead to therapeutic overdose due to the low levels of antithrombin III (the substrate to which LMWH binds) in patients with advanced disease. Thus, anti-Xa levels were found to be significantly lower in patients with cirrhosis despite an adequate anticoagulation effect [135,136]. Despite this limitation, we still use anti-Xa levels to help us adjust the dose in patients with bleeding complications on LMWH therapy, more so in the presence of renal dysfunction and morbid obesity.

#### 10.2.3. Fondaparinux

Fondaparinux constitutes a molecule that inhibits activated factor X through selective high-affinity binding to antithrombin III [137]. Unlike heparin, fondaparinux does not inhibit thrombin directly or platelet factor IV. For this reason, risk of heparin-induced thrombocytopenia is much lower [138]. Its once-daily administration makes it more convenient than most LMWH.

Regarding its efficacy and safety, fondaparinux was compared to LMWH in one retrospective study including 124 patients with cirrhosis and PVT. Fondaparinux showed higher probability of resolution of PVT at 36 months (77% vs. 51%; *p* = 0.001) but also higher bleeding rates (27% vs. 13%; *p* = 0.06) [139]. Further prospective studies are required before making a formal recommendation.

#### 10.2.4. Vitamin K Antagonists

The use of VKAs with the aforementioned “bridge” strategy was shown to be safe and effective in both waitlist and non-waitlist cohorts [102,126]. Recanalization rates and adverse effects seem to be similar when compared to LMWH [104,120]. The main limitations of VKA are their narrow therapeutic window and the risk of underdosing due to the baseline elevation of INR observed in patients with advanced cirrhosis.

#### 10.2.5. Direct Oral Anticoagulants

DOACs, including direct thrombin inhibitor (dabigatran) and factor Xa inhibitors (rivaroxaban, apixaban, and edoxaban) were recently added to the PVT therapeutic armamentarium. Their main advantages are their oral administration in fixed doses and poor interaction with other drugs. Current available data suggest that there are no major safety concerns regarding the use of DOACs in patients with Child–Pugh A and that they should be used with caution in patients with Child–Pugh B (some authors do not recommend rivaroxaban in these patients because of increased plasma concentrations and pharmacodynamic effects [140]). In patients with Child–Pugh C, DOACs are not recommended [4]

In regard to their pharmacokinetics in cirrhosis, very little is known. In vitro studies using plasma of decompensated cirrhosis have shown differences in anticoagulation potency when measured by thrombin generation assays [141,142]. Defects in various steps of drug metabolism such as plasma protein binding, cytochrome p450 function, biliary excretion, and renal clearance due to the underlying cirrhosis may partially explain these results, although larger in vivo studies are needed [3,84].

Several retrospective studies [143,144,145,146,147] and meta-analyses [148,149] have observed a similar or better safety profile in terms of bleeding complications of DOACs compared to VKAs in cirrhotic patients affected by atrial fibrillation, venous thromboembolism, or PVT. Bleeding definitions differed among these studies, and to control for this bias, Nisly et al. conducted a systematic review and meta-analysis considering only studies in which the primary safety outcome was major bleeding according to the definition of the International Society on Thrombosis and Haemostasis. They found no major differences in this regard between DOACS and traditional anticoagulants [148]. It must be noted, however, that the bleeding risk increased in patients with advanced cirrhosis. In a retrospective study, Semmler et al. observed a significant association of spontaneous bleedings with liver disease severity [147].

There is even less evidence in relation to the effectiveness of DOACs in patients with cirrhosis and PVT. In the only randomized, controlled trial, Hanafy et al. showed that rivaroxaban was more effective than warfarin in terms of recanalization rates, recurrence of PVT, and safety in patients with HCV-related cirrhosis and PVT [150]. Similar findings were found in a retrospective study comparing edoxaban with warfarin [151] and in a prospective study comparing ribaroxaban and dabigatran with no treatment [152].

Due to this increasing evidence, their much easier administration, and the existence of reversal agents (idarucizumab for dabigatran and andexanet alfa for rivaroxaban and apixaban), these agents are increasingly utilized. Current barriers, beyond safety issues in patients with advanced liver disease, are their high cost and limited availability.

### 10.3. Transjugular Intrahepatic Porto-Systemic Shunt

PVT was previously considered a contraindication to TIPS creation. However, the introduction of new interventional radiological techniques has improved the rates of technical success up to 86.7–95%. It must be highlighted that most of the available data come from small, non-controlled, and retrospective cohorts that differ in inclusion criteria and treatment applied. Moreover, complications of portal hypertension refractory to conventional therapy, and not PVT itself, were the most common indications for TIPS placement in patients with cirrhosis and PVT [153,154,155].

Two meta-analyses have shown recanalization rates after TIPS of 81–84.4% and of 73% for complete recanalization. The rates of major complications were 10% [156,157]. When compared to anticoagulation, TIPS has a higher effectiveness in terms of portal vein recanalization [158,159]. Regarding portal-hypertension-related complications, two randomized controlled trials have evaluated the effectiveness of TIPS in comparison to standard therapy (endoscopic band ligation + propranolol + anticoagulation) for the prevention of variceal rebleeding in cirrhotic patients with PVT. Both studies showed that TIPS was more effective in preventing rebleeding and in recanalizing PVT without increasing the risk of hepatic encephalopathy and adverse effects. These beneficial effects, however, did not translate into improved survival [160,161]. Both of these trials used anticoagulation after TIPS placement, although indications of anticoagulation after TIPS placement are not clear. In fact, current evidence does not support its use for further improving recanalization rates, although it might be considered in cases of PVT extension to the SMV [32,156,158,162].

Patients with portal cavernoma or with no identifiable intrahepatic portal trunk or branches are the most challenging. For these patients, a modified transplenic or transhepatic approach, known as portal vein recanalization-TIPS, was shown to improve technical success to over 90% in the most recent series (range 75–100%) [163,164,165,166]. In these procedures, portal vein recanalization is performed by angioplasty/stenting with subsequent TIPS insertion to ensure the outflow of the system. TIPS failure may be due to lack of a landing zone at the distal end of the portal vein or at the spleno-mesenteric confluence [167].

In line with the above, current guidelines recommend considering TIPS for the following indications in patients with cirrhosis and PVT: (1) inadequate response to or contraindication of anticoagulation; (2) chronic PVT/portal cavernoma with portal-hypertension-related complications refractory to medical treatment; and (3) chronic PVT that hinders a physiological anastomosis between the graft and recipient portal vein [1,2,3,4,168]. Figure 5 shows the therapeutic algorithm according to the latest Baveno VII consensus [4].

### 10.4. Thrombolysis

Local or systemic thrombolysis was proposed as an adjunct to anticoagulation for the treatment of recent PVT. Small case series have reported similar efficacy rates to those achieved with anticoagulation alone but with a high risk of procedure-related morbidity and mortality [169,170,171]. Based on these preliminary observations, current guidelines only recommend thrombolitic therapy in specialized centers for very selected patients in whom intestinal ischemia persists despite anticoagulation [3,12].

### 10.5. Other Considerations

The management of complications of portal hypertension should not differ from those of other patients with cirrhosis. Anticoagulation must be started always after implementing an adequate prophylaxis for gastrointestinal bleeding [1]. Therefore, decompensated patients not receiving NSBB should undergo screening endoscopy while patients with compensated advanced chronic liver disease with unequivocal signs of clinically significant portal hypertension may be started directly on NSBB. Otherwise, an upper endoscopy should be performed in patients not fulfilling the Baveno VI criteria [4]. If these criteria are met, this decision should be individualized depending on the age and extent of PVT and on whether anticoagulant therapy is to be initiated. Importantly, recent data suggest that anticoagulation should not be delayed until variceal eradication or complete β-blockade is achieved [3]. This recommendation also applies to the setting of secondary prophylaxis. Indeed, the continuation of LMWH through prophylactic endoscopic variceal ligation did not increase the risk of bleeding in patients with cirrhosis [172], with similar findings in patients with non-cirrhotic PVT [173]. Further data in this regard are needed.

Patients with HCC invasion of the portal vein do not benefit from anticoagulation [2]. However, in some rare cases, the initial “malignant” thrombus induces a stagnant flow that can lead to clot formation and symptomatic PVT (e.g., intestinal ischemia). In this infrequent scenario, anticoagulation may be considered after weighing the risk and benefits. In regard to cancer-associated symptomatic recent (non-cirrhotic) splanchnic vein thrombosis, the International Society on Thrombosis and Haemostasis recommends LMWH or DOACs. They suggest LMWH in patients with luminal gastrointestinal cancer, active gastrointestinal mucosal abnormalities, genitourinary cancer at high risk of bleeding, or receiving current systemic therapy with potentially relevant drug–drug interactions with DOACs [12].

Finally, we have a few words regarding the prevention of PVT after LT. Pre-LT PVT is a risk factor for PVT recurrence, with a greater risk if a non-anatomical anastomosis is performed or if pre-LT PVT was of a great degree and extension [98]. In the recent consensus of the Spanish Society of Liver Transplantation and the Spanish Society of Thrombosis and Hemostasis, therapeutic LMWH (i.e., 1 mg/kg) started within the first 24 h after surgery is recommended in patients with risk factors of PVT in the absence of coagulopathy, liver graft dysfunction, or low platelet count (<30,000–50,000/μL). Risk factors include pre-LT PVT, slow portal flow (after reperfusion) defined as <1300 mL/min or <65 mL/min/100 g, partial thrombectomy or vein intimal layer lesion during thrombectomy, nonphysiologically portal vein inflow reconstruction, and thrombophilic disorders in the recipient. In the absence of complications, therapy should be prolonged at least 2 months after LT and individualized thereafter [174].

## 11. Conclusions

Portal vein thrombosis constitutes the most common thrombotic event in patients with cirrhosis. The recent incorporation of DOACs and of new radiological techniques for portal vein recanalization are major breakthroughs in the treatment of this complication. However, the lack of large prospective observational studies and randomized trials explain the uncertainties regarding its impact on the natural history of cirrhosis and the heterogenous diagnostic and therapeutic recommendations of current guidelines. Thus, future research to fill current gaps of knowledge is needed and will likely require multicenter collaboration.

## Figures and Tables

**Figure 1 jcm-11-06435-f001:**
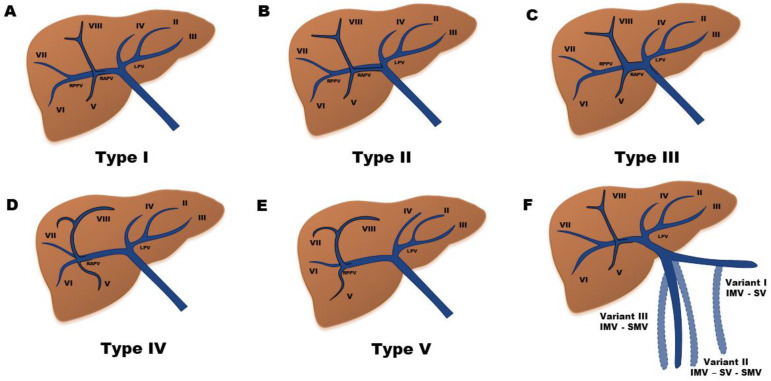
The intrahepatic portal vein has a segmental distribution and it closely follows the hepatic artery. Cheng Y [15] et al. classified intrahepatic portal vein variations into five different types. (**A**). Type 1 (65–80%): after its entry through the hilium, the main portal vein (MPV) divides into a larger right portal vein (RPV) and a smaller left portal vein (LPV). RPV is divided into an anterior branch (supplying segments V and VIII) and a posterior branch (supplying segments VI and VII). LVP runs horizontally to the left and then turns medially (supplying segments I, II, III, and IV). (**B**). Type II (10–15%): trifurcation of MVP, dividing into the right anterior and posterior branches and the LPV. (**C**). Type III (0.3–7%): the right posterior portal branch arises directly from the MPV as its first branch and the LPV is the terminal branch, arising after the origin of the right anterior portal vein. (**D**). Type IV (0.6–2.7%): trifurcation of the RPV, in which the branch of segment VII is the first branch of the RPV. (**E**). Type V (1.3–2.4%): trifurcation of the RPV, in which the branch of segment VI arises early as a separate branch of the RPV. (**F**). Different variants in the drainage of the inferior mesenteric vein (IMV). I: drainage into the splenic vein (SV). II: drainage into the confluence of the superior mesenteric vein (SMV) and SV. III: drainage into the SMV.

**Figure 2 jcm-11-06435-f002:**
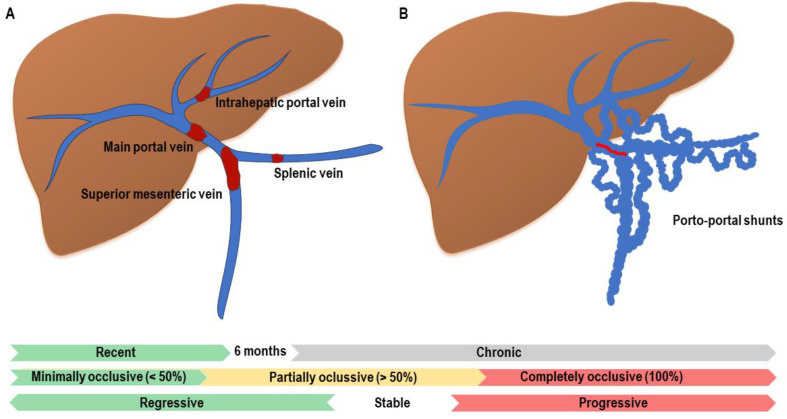
Classification of portal vein thrombosis. (**A**). It is crucial to characterize the extension of the portal vein thrombosis and its relationship with the main portal vein (MPV). The involvement of the superior mesenteric vein (SMV) and splenic vein (SV) should be properly characterized due to its implication in prognosis and treatment. (**B**). Cavernous transformation of the portal vein. Atrophy of the MPV (marked in red) secondary to chronic PVT causes hypertrophy in the vasa vasorum of the portal vein with the apparition of porto-portal shunts.

**Figure 3 jcm-11-06435-f003:**
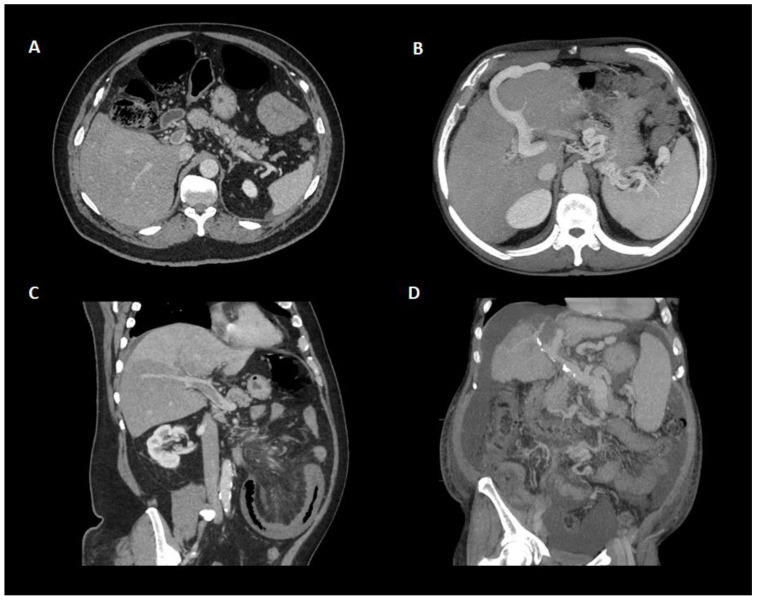
Different presentations of portal vein thrombosis through contrast-enhanced CT scan. (**A**) Axial plane showing a partially occlusive portal vein thrombosis of the main portal trunk without collaterals or other indirect signs of portal hypertension. (**B**) Apparition of collateral circulation with umbilical vein repermeabilization and enlarged spleen in the context of portal vein thrombosis. (**C**) Oblique plane showing a partially occlusive portal vein thrombosis of the main portal trunk with hepatic perfusion defects on the lower segments of the right lobe. (**D**) Chronic portal vein thrombosis in patient with cirrhosis. Note the calcification of the portal vein as an indirect sign of chronicity. Atrophic liver with splenomegaly, collaterals, and ascites are found in the context of cirrhosis.

**Figure 4 jcm-11-06435-f004:**
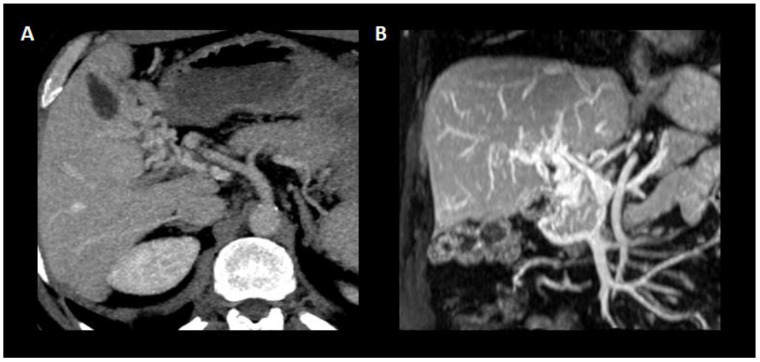
Cavernous transformation of the portal vein. (**A**) Axial plane showing the substitution of the main portal vein trunk by porto-portal collaterals originated from the vasa vasorum of the portal vein. (**B**) Coronal plane of the portal vein and its cavernous transformation with the entanglement of the new vessels composing the porto-portal collaterals.

**Figure 5 jcm-11-06435-f005:**
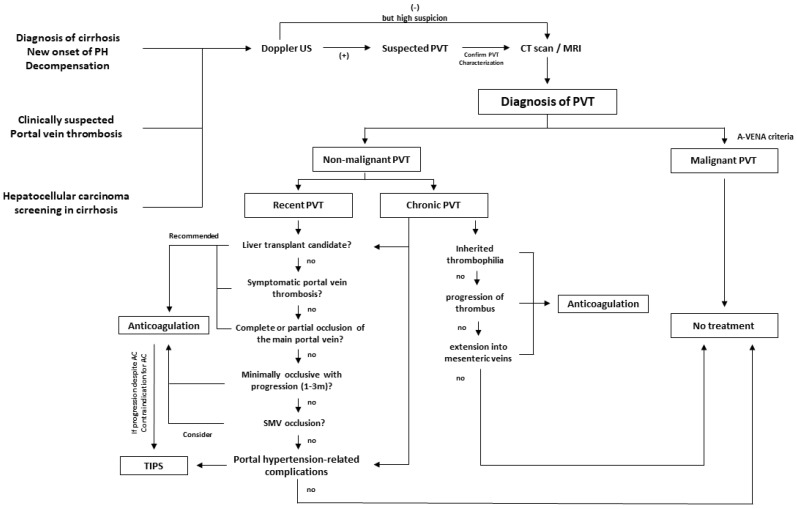
The algorithm for the management of portal vein thrombosis in patients with cirrhosis according to the Baveno VII consensus recommendations.

**Table 1 jcm-11-06435-t001:** Classification of portal vein thrombosis according several authors.

Classification	Description	Categories	Strengths vs. Weaknesses
Yerdel et al., 2000 [16]	Post-transplant survival (i) Involvement of MPV, SMV and SV (ii) Degree of occlusion	Grade 1: <50% occlusion Grade 2: >50% occlusion Grade 3: Complete PV or SV occlusionGrade 4: Complete PV occlusion with SMV extension	- Well-defined long-term post-transplant survival - Only for LT candidates - Most widely used classification to date
Bauer et al., 2006 [17]	PVT response to TIPS placement in patients listed for LT (i) Involvement of MPV, SMV and SV (ii) Degree of occlusion (iii) Stratification by location of clot and cavernous transformation	Grade 1: <25% occlusion of PV Grade 2: 26–50% occlusion Grade 3: 51–75% occlusion Grade 4: 76–100% occlusion	- Available data not only in LT recipients but also in non-LT patients and patients with cavernous transformation - Better characterization of the location and degree of occlusion than other classifications - Only for patients who underwent TIPS placement. Validated in 9 patients. No long-term outcome data.
Sarin et al., 2016 [18]	Based on revision of previous PVT classifications (i) Involvement of MPV, SMV, and SV (ii) Degree of occlusion (iii) Underlying liver disease	Site of PVT (Type 1, 2a, 2b, 3) * Type 1: only trunk * Type II: only branches (a: one, b: both branches) * Type 3: trunk and branches Degree of occlusion * Occlusive * Non-occlusive Duration and presentation * Recent or chronic * Symptomatic or asymptomatic Extension * SV, SMV, or both Underlying liver disease	- “Anatomical-functional” classification - Complex - No correlation with outcomes
Bhangui et al., 2019 [19]	(i) Involvement of MPV, SMV and SV (ii) LT surgery choices based on collaterals	Complex - SYerdel grade 4 and Jamieson and Charco grades 3/4 - Non-complex - Yerdel grade 1–3	- Simplifies anatomical considerations for PV management. - Mainly focused on LT surgical techniques not in outcomes.

Abbreviations: LT: liver transplant; MPV: main portal vein; PH: portal hypertension; PV: portal vein; PVT: portal vein thrombosis; SMV: superior mesenteric vein; SV: splenic vein; TIPS: transjugular intrahepatic portosystemic shunt.

**Table 2 jcm-11-06435-t002:** Current guidelines recommendations for management of portal vein thrombosis in cirrhosis.

	Baveno VII 2022	AASLD 2021	ACG 2020	ICLDC 2018	EASL 2015
**Screening for PVT**	Listed or potential candidates for LT at the time of HCC screening	-	(i) new diagnosis of cirrhosis (ii) onset of PH (iii) decompensation.	US every 6 months in (i) patients with cirrhosis and PH, or (ii) Listed or potential candidates for LT	Listed or potential candidates for LT
**Imaging test**	- Screening: US - Diagnosis: CT scan or MRI	- Diagnosis: CT scan or MRI	- Screening: US - Diagnosis: CT scan or MRI	- Screening: US - Diagnosis: CT scan or MRI	- Diagnosis: CT scan or MRI
**Screening for thrombophilia**	-	(i) family history of MPN (ii) suggestive laboratory findings	(i) previous thrombosis (ii) thrombosis at unusual sites (iii) family history	“Consider on an individual basis”	“Consider screening”
**Indications for treatment**	- Recommended: (i) recent (<6 m) PVT completely or partially occlusive of the PV trunk (ii) symptomatic PVT (iii) potential candidates for LT - Considered: (iv) minimally occlusive that progresses (v) compromise of SMV	(i) Recent (<6 m) completely or partially occlusive of the main PV or SMV (ii) Ischemic symptoms	(i) evidence of thrombophilia, (ii) progression into the mesenteric veins, or (iii) current or previous evidence of bowel ischemia	(i) LT candidates with occlusive main PVT with or without extension to SMV (ii) Yerdel Grade ≥ 2 PVT considered on an individual basis	(i) SMV thrombosis, with a past history suggestive of intestinal ischemia or (ii) liver transplant candidates
**Medical therapies**	Initial agent: preferably LWMH Maintenance: - LWMH, VKAs, DOACs	LWMH, VKAs or DOACs	Initial agent: UH or LWMH Maintenance: L - WMH, VKAs	-	-
**TIPS**	(i) PVT of the main PV without recanalization on AC, especially in patients listed for LT	(i) PVT that hinders a physiological anastomosis between the graft and recipient (ii) refractary PH complications	-	(i) acute and chronic PVT in patients with cirrhosis requiring treatment for significant PH	(i) LT candidates not responding to AC

Abbreviations: AASLD: American Association for the Study of Liver Diseases; AC: anticoagulant; ACG: American College of Gastroenterology; DOACs: direct oral anticoagulants; EASL: European Association for the Study of the Liver; HCC: hepatocellular carcinoma; ICLDC: International Coagulation in Liver Diseases Conference; LT: liver transplant; LWMH: low weight molecular heparin; MPN: myeloproliferative neoplasm; MRI: magnetic resonance imaging; SMV: superior mesenteric vein; SV: splenic vein; TIPS: transjugular intrahepatic portosystemic shunt; PH: portal hypertension; PV: portal vein; PVT: portal vein thrombosis; UH: unfractionated heparin; US: ultrasound; VKAs: vitamin K antagonists.

**Table 3 jcm-11-06435-t003:** Characteristics of the different anticoagulation therapies for the treatment of portal vein thrombosis.

	Unfractionated Heparin	Low-Weight Molecular Heparin	Vitamin-K Antagonists	Direct Oral Anticoagulants
**Administration**	Endovenous	Subcutaneous	Oral	Oral
**Posology**	Daily infusion	qd/bid	Qd	Apixaban/dabigatran: bid Edoxaban: qd Rivaroxaban: bd for 3 weeks, qd thereafter
**Half-life**	Minutes to 1–2 h	4–12 h	10–24 h	6–18 h
**Absorption and bioavailability**	Caution if hypoalbuminemia	BA 85–95%, caution if hypoalbuminemia	Affected from bowel edema in PH and diet	Affected from bowel edema in PH
**Monitoring**	antiXa factor or aPTT	Not needed, but caution if GFR < 15 mL/min/m^2^, obesity, and female sex	PT and INR (2–3)	Not needed, especially if GFR > 15 mL/min/m^2^, non-obese, and male sex
**Renal function**	Dose change not necessary but monitor with antiXa and aPTT	Contraindicated if severe renal failure or dialysis, caution in mild/moderate renal failure	Contraindicated if severe renal failure or dialysis, caution in mild/moderate renal failure	- No dose change needed - Not indicated if GFR < 15 mL/min/m^2^
**Side Effects**	- Hemorrhage/hematoma - Heparin-induced thrombopenia (+++) - Hyperkalemia	- Hemorrhage/hematoma - Heparin-induced thrombopenia (++) - Altered LFT	Hemorrhage/hematoma	- Hemorrhage/hematoma - Altered LFT
**Bleeding risk**	++	++	++	Apixaban: + Dabigatran/Edoxaban: ++ Rivaroxaban: +++
**Antidote**	Protamine sulfate	Protamine sulfate *	Vitamin-K	AntiX: andexanet alpha ** Dabigatran: idarucizumab **
**Pros in cirrhosis**	- Allowed in renal failure - Short half-life	- Subcutaneous administration - Easy interruption before invasive procedures	Oral administration	- Oral administration - Predictable effect - Fewer interactions than VKAs
**Cons in cirrhosis**	Not suitable for maintainance therapy Fluctuating levels of ATIII	- Renal function lability - Not well monitored through antiXa	- INR not reliable in cirrhosis	Not recommended in Child C Caution in Child B Higher costs

Abbreviations; AC: anticoagulation; antiX: apixaban, edoxaban, and rivaroxaban; aPPT: activated-partial thromboplastin clotting time; ATIII: antithrombin III; BA: bioavailability; DOACs: direct oral anticoagulants; GFR: glomerular filtration rate; LFT: liver function test; LWMH: low-weight molecular heparin; PH: portal hypertension; PT: prothrombin time; INR: international normalized ratio; UH: unfractionated heparin; VKAs: vitamin-K antagonists. * Less effective in LWMH than unfractionated heparin. ** Available only in selected centers.

## Data Availability

Not applicable.

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
