# Peer review of "Portal Vein Thrombosis in the Setting of Cirrhosis: A Comprehensive Review"

_jcm, 2022, doi:10.3390/jcm11216435_

Round 1

Reviewer 1 Report

This is a comprehensive review on PVT in patients with cirrhosis. This paper is well-written and presented, and I think that the Authors should be congratulated for their efforts. 

I have no major issue with this work. My only suggestion for the Authors would be to include and discuss some additional, recent findings on PVT in patients with cirrhosis such as:

1) Pathophysiology of PVT. There are recent data indicating that platelets are hyper-functional in patients with cirrhosis (10.1016/j.jhep.2022.03.009), especially in the portal vein (10.1080/09537104.2022.2060499). In fact, a recent prospective study showed that increased platelet aggregation is independently associated with PVT in decompensated patients (https://doi.org/10.1111/liv.15435). Similarly, an independent cohort showed that the ratio between VWF/ADAMTS13 is predictive of PVT (10.1016/j.dld.2022.06.004). These results could be useful for prediction of PVT and open the discussion regarding the need for further studies to investigate the use of antiplatelet agents to prevent PVT in cirrhosis and would be of interest in such a review.

2) Patients with cirrhosis who develop HCC may be at increased risk of PVT, which may be due to multiple alterations of hemostasis driven by HCC (see 10.1111/liv.15183). It may be worthy to include a brief statement regarding this association to make the clinicians aware that these patients may be at higher risk. Since patients with HCC may be candidate for transplantation, and since PVT is associated with increased risk of post-transplant mortality (10.1111/tri.13353), these patients may deserve special consideration.

3) Recent evidence indicates that, in acute decompensation of cirrhosis, systemic inflammation is the main driver not only for development of ACLF, but also bleeding and thrombosis. A level of PAI-1 greater than 50 has been associated with a much higher risk of developing PVT and non-splanchnic venous thrombosis (10.1016/j.jhep.2022.09.005). Since – if confirmed by further studies, this could be a simple biomarker for identification of patients at higher risk of venous thrombosis, I think it may be interesting to include these results that would enrich the review.

Author Response

We would like to sincerely thank the reviewers for their kind comments and constructive criticisms. We hope to have adequately addressed them. We now explain point-by-point the details of the revisions in the manuscript and our responses to the reviewers' comments. All the corrections/suggestions added in response to the reviewers´ comments have been highlighted in green in the manuscript. We have also corrected spelling mistakes. These have been highlighted in yellow

Reviewer 1:

We would like to sincerely thank the reviewer for his kind comments and suggestions. We have added in section 4.2 (pages 6 and 7) all the suggestions of the reviewer. It now reads as follows (changes are highlighted in green):

“Several studies have evaluated whether the hemostatic alterations associated with cirrhosis increase the risk of PVT and have been extensively reviewed elsewhere (24). Traditional coagulation tests (e.g., prothrombin time or partial thromboplastin time) do not adequately reflect this new hemostatic balance, since they do not take into account the inhibition of thrombin by anticoagulant factors (3, 36). The ratio between FVIII and protein C has long been suggested to reflect this increase in coagulation potential (38), although recent data has challenged this assumption showing that despite predicting the devel-opment of complications of cirrhosis, it is unrelated to the coagulation status of patients with cirrhosis (39). Regarding its role in PVT, contradictory results have been published (28, 40, 41). Similar findings have been observed with other cirrhosis-related hemostatic alterations such as the ratio of factor II to protein C, levels of coagulation factors, throm-bomodulin resistance, fibrinolysis markers, plasminogen activator inhibitor-1 levels (42)or viscoelastic parameters (24). Of note, the aforementioned studies evaluated coagulation factors in systemic blood with few studies evaluating whether the portal vein may rep-resent a hypercoagulable vascular bed. Although initial studies described the existence of a relative hypercoagulability in this territory (43, 44), a more recent study failed to rep-licate this finding in cirrhotic patients who underwent TIPS placement (45). Its hypo-thetical contribution in PVT development has not been tested as none of these studies evaluated patients with PVT.

There is less data on the role of platelet aggregation in PVT development. In op-position to previous studies suggesting platelet dysfunction in patients with cirrhosis, recent studies show that platelets are hyperfunctional in these patients, particularly in the decompensated stage and in the portal vein (46-48). Moreover, this increased platelet aggregatory potential has been associated with a higher risk of further decompensation, death, and PVT (47, 49). In line with these findings, a recent study showed that the ADAMTS-13/von Willebrand factor ratio was predictive of PVT (50). These results es-tablish a rational basis for evaluating the use of antiplatelet agents to prevent PVT and halt disease progression.

Other factors that may induce blood hypercoagulability in patients with cirrhosis is systemic inflammation, a well-recognized feature of decompensated cirrhosis (51), and HCC (52). The former has been scarcely studied with contradictory results (24, 28). A recent paper observed that serum albumin was inversely associated with PVT and suggested albumin as a modulator of the hemostatic system by reducing platelet acti-vation through its inhibitory effects on oxidative stress. According to the authors, these findings established a rationale for randomized interventional studies to investigate the beneficial effects of albumin to prevent PVT in cirrhosis (53). No information in this matter has been provided in the long-term albumin administration trials (54, 55). As far as HCC is concerned, there is growing evidence suggesting that it is associated with pro-thrombotic alterations (i.e., increased platelet activation and function, enhanced thrombin generation, hypo-fibrinolysis and elevated levels of prothrombotic microvesi-cles) that may synergistically contribute to hypercoagulability and thrombosis (52)."

Reviewer 2 Report

This is a very comprehensive review of the available information on PVT. I have only some recommendations. Altogether the review is a bit long. The authors should identify areas that can be shortened or cut.

Please state which is the most common classification for PVT

Figure 3 did you mean contrast enhanced CT scan?

Correct the typo Fondaparinaux

Correct format in Conclusion

Author Response

We would like to sincerely thank the reviewers for their kind comments and constructive criticisms. We hope to have adequately addressed them. We now explain point-by-point the details of the revisions in the manuscript and our responses to the reviewers' comments. All the corrections/suggestions added in response to the reviewers´ comments have been highlighted in green in the manuscript. We have also corrected spelling mistakes. These have been highlighted in yellow

Reviewer 2:

This is a very comprehensive review of the available information on PVT. I have only some recommendations. Altogether the review is a bit long. The authors should identify areas that can be shortened or cut.

We would like to thank the reviewer for his kind comments. When we first designed this manuscript, we wanted to make a comprehensive review to allow readers to have a broad overview of this complication. Many previous reviews focus only on certain aspects of PVT, and our goal was to overcome this at the expense of a longer manuscript. We apologize for not shortening or cutting any part as the reviewer suggested.

Please state which is the most common classification for PVT.

Until now, it is the classification of Yerdel. We have added this statement in Table 1.

Figure 3 did you mean contrast enhanced CT scan?

Yes, we apologize for this error that has been corrected.

Correct the typo Fondaparinaux

We apologize for this error that has been corrected.

Correct format in Conclusion

We understand that the reviewer refers to the fact that the paragraph was misplaced. We have corrected this error.

Round 2

Reviewer 1 Report

The authors have addressed all my comments and suggestions.